# Anticipatory and pre-planned actions: A comparison between young soccer players and swimmers

**Francesca Nardello, Matteo Bertucco, Paola Cesari** *

Department of Neurosciences, Biomedicine and Movement Sciences, University of Verona, Verona, Italy

* paola.cesari@univr.it

**Data Availability Statement:** All relevant data are within the paper and its Supporting Information files.

**Funding:** The authors received no specific funding for this work.

## Abstract

The present study investigated whether a difference exists in reactive and proactive control for sport considered open or closed skills dominated. Sixteen young (11–12 years) athletes (eight soccer players and eight swimmers) were asked to be engaged into two games competitions that required either a reactive and a proactive type of control. By means of kinematic (i.e. movement time and duration) and dynamic analysis through the force platform (i.e. Anticipatory Postural Adjustments, APAs), we evaluated the level of ability and stability in reacting and anticipating actions. Results indicated that soccer players outperformed swimmers by showing higher stability and a smaller number of falls during the competition where proactive control was mainly required. Soccer players were able to reach that result by anticipating actions through well-modulated APAs. On the contrary, during the competition where reactive control was mainly required, performances were comparable between groups. Therefore, the development of specific action control is already established at 11–12 years of age and is enhanced by the training specificity.

## Introduction

Sports are traditionally classified in two general categories based on the relationship between the movements and the environmental changes allowed by the task at hand [1]. The categories are recognized as open and closed skills: open skills are those that take place in an unpredictable and constantly changing environment where movements need to be continually adapted (i.e. basketball, soccer, volleyball, handball); vice versa closed-skills are performed in an unchanged environment with movements that follow a relatively stable set of patterns as track and field or swimming [2–4]. Consider for instance the game of basketball where the ball and the players are constantly moving. In this case, it is fundamental to develop specific abilities as to adapt dynamically to the changes of the game contexts or to anticipate the actions performed by others and predict the trajectories of the ball [4, 5]. This is not the case for a swimmer; he/she is not asked to react to the other swimmers' actions nor to anticipate unknown outcomes, but instead to maintain core stability while performing in the most efficient way repeated patterns of action [6]. It turns out that the ability to deal with the relationship between

**Competing interests:** The authors have declared that no competing interests exist.

environmental contexts and movement responses for an open skilled athlete could be different than the one obtained by a closed skilled type of athlete [7]. Nevertheless, elite athletes whatever the sport practiced, develop common sensory-motor capabilities.

Sports are performed under conditions of stress due to the physical, psychological and environmental demands, for satisfying expectations and resisting to pressure while performing high level executions [8]. Such conditions exacerbate the athlete's ability to quickly and accurately pick up relevant information for the task at hand; this learning process would reduce the time of making a decision and allow more accurate action preparation [9, 10]. In addition, either in closed or open dominated-sport skills, fast reaction time (RT) underlines the presence of critical abilities considered advantageous to the player's successful performance [11]. In fact, in many sports, maximum speed is rarely reached or needed, while instead explosive reaction is often required [12]. Reacting faster to a stimulus has been considered a basic measure to assess many different sensory motor capacities, from simple daily life tasks [13], to high competitions as in karate [14] and athletic dash sprint [15].

Following behavioural results [16], researches asked what control will be necessary for developing open or closed type of motor skill by searching for differences between "proactive" and "reactive" control of movement through the disclosure of possible related neuronal underpinning [17]. Proactive control is mainly defined by a process that allows an early selection of the response helped by an appropriate attention and cue selection along with a goal-driven action [18]. This action control mechanism is applied as an essential knowledge for understanding the future of an action for being able to anticipate the action. Overall, this type of control is more present in an open environment [4]. On the contrary, reactive control requires to respond imperatively after the stimulus appear [18]. In this case a fast and appropriate reaction of external changes represents the typical control present in a closed environment [19]. The two controls require different neurophysiological processes: the proactive being more driven by a feedforward control while the reactive more by a feedback control [20].

One way to investigate the link between action planning (feedforward control) and execution (feedback control) is to measure the muscles activity before movement initiation, since voluntary actions are always preceded by postural changes [21, 22]. These changes occur prior to the movement itself and can be conceived of as Anticipatory Postural Adjustments (APAs). APAs are centrally programmed and their putative role is to minimize perturbations to vertical posture that would otherwise be induced by a movement (for a review see [23]). Starting from early pioneering research, APAs have been studied during lower limb movement [24], trunk movement and arm movement [25]. In literature, APAs are usually studied with through EMG analysis (i.e. early postural adjustments, EPA) [22, 25] and/or with force platform [24, 26–28]. APAs are typically quantified in terms of their magnitude or the onset timing of the signal (i.e. EMG or GRFs) before the initiation of the movement [26, 27]. When measured with the ground reaction force [29–31], APAs onset, amplitude and impulse are usually analysed from the ground reaction forces (GRFs) [24, 26–27] or centre of pressure [28].

To our knowledge, this is the first study that measures anticipatory mechanisms during game competition for comparing young but highly expert athletes that developed abilities in a more dynamic or in a more stable environment. Past literature has investigated the superiority of the skilled athletes over less skilled athletes [16, 32], but only a limited number of studies have directly compared anticipatory skills performed by athletes from closed skill-dominated and open skill-dominated sports [32, 33] and mostly considering adult athletes. Critically, these studies considered as a test mainly tasks to be performed on a computer table game situation as for instance the "task-switching paradigm" [17, 34]. Therefore, given the scarce literature about the development of proactive and reactive control mechanisms and the absence of test that considers more ecologically how a real body game situation let merges reactive and

proactive skills, we want to analyse the behaviour of APAs by comparing the anticipatory skills of soccer players (open skill-dominated sport) and swimmers (closed skill-dominated sport). This will reveal how the CNS control action anticipation in young athletes when engaged in a total body game competition requiring proactive and reactive control. The importance to test expert children is to better understand the development of the appearance of these anticipatory mechanisms indicating the influence that a sport training has in enhancing specific motor skills.

The main aim of this study is to shade light on the role of skills development (open and closed) by comparing APAs, by means of the ground reaction forces, in children highly experienced in open and closed disciplines respectively soccer and swim.

In a proactive game, we would expect a greater postural stability accomplished with a lower risk of falling for the open-skills players in comparison to the closed-skills ones. This augmented ability would be reflected by an early modulation of the Anticipatory Postural Adjustments. In a reactive game, instead, we assume no differences between open and closed skills.

## Material and methods

### Participants

Sixteen young male athletes were recruited for the experiment: eight were soccer players and eight swimmers. They were recruited from the A.C. Chievo Verona soccer club and from the CSS "Montebianco" swimming pool club in Verona, respectively. Their anthropometric characteristics and years of experience are reported in Table 1.

All participants were free from any musculoskeletal disorder and neurologic disease that could affect the study. The project was approved by the Ethical Committee at the Department of Neurosciences, Biomedicine and Movement Sciences at University of Verona (Prot. 269664/2017) and all participants provided written informed consent before taking part in the experimental procedures.

### Apparatus

Video clips were recorded by means of a digital camera (Panasonic, Lumix DMC FZ200, made in China, 100Hz, 12.1Megapixel, optical zoom 24x). The camera was located perpendicular to the sagittal plane of the game zone, on the left side of the players.

Two force platforms (model OR-5, AMTI, USA: $90 \times 90$ cm and model Kistler, Switzerland: $40 \times 60$ cm) were used to record the three components of the ground reaction forces with a sampling rate of 2000 Hz. Only the anterior-posterior component (GRFx) has been used for further analysis.

A couple of pads was attached on both palms for each subject with the Velcro®. The aluminium-made pads were covered with three layers of felt and externally with leather to soften the impact. One tri-axial accelerometer (model LSM6DS33TR, Farnell, ITALY, sampling rate

**Table 1. Soccer players and swimmers group body characteristics (mean (SD)).**

| Variable | Soccer players | Swimmers |
|---|---|---|
|  | (*n* = 8) | (*n* = 8) |
| Age (years) | 11.8 (0.3) | 11.6 (0.6) |
| Body Height (m) | 1.54 (0.06) | 1.54 (0.09) |
| Body Mass (kg) | 44.7 (6.11) | 43.7 (6.70) |
| Years of experience | 3.8 (0.3) | 3.6 (0.5) |
| Training hours for week | 6.00 | 6.00 |

2000 Hz) was attached internally to the pad which was tied to the player's dominant arm. The signals were used to quantify the movement time up to the impact.

## Experimental procedure

**Proactive type of game.** In order to test action anticipation ability, subjects were asked to participate to the "push and fall" game. The game is played by a couple of individuals (player 1 *versus* player 2); each couple was formed by a soccer player (e.g. player 1) and a swimmer (e.g. player 2). The two players were standing barefoot on a force plate each, facing each other (Fig 1, panel a). As initial position they were asked to stand with feet together, to flex their upper arm by facing their palms to the opponent. The distance between the two players corresponded to the length of their arms as to be able to touch the shoulder of the opponent with their arm extended forward.

The main rule of the game was to push with palm contact the opponent palms to threw him off balance. The trial could be initiated after an external vocal command "push" was delivered. After the push command players were free to decide when to initiate the action. No constraints for time initiation of the movement was given. Each player was allowed to move just in one shot (i.e. a single push). In other words, fake actions were not allowed. More in specific extra movements for simulating fake actions were not allowed but players could choose not to move at all so that in some trials impact between the two players' palms was not present. The contact between the two players was allowed only by touching the palms; no other parts of the body could come into contact. The players were not asked to react as soon as possible to the push

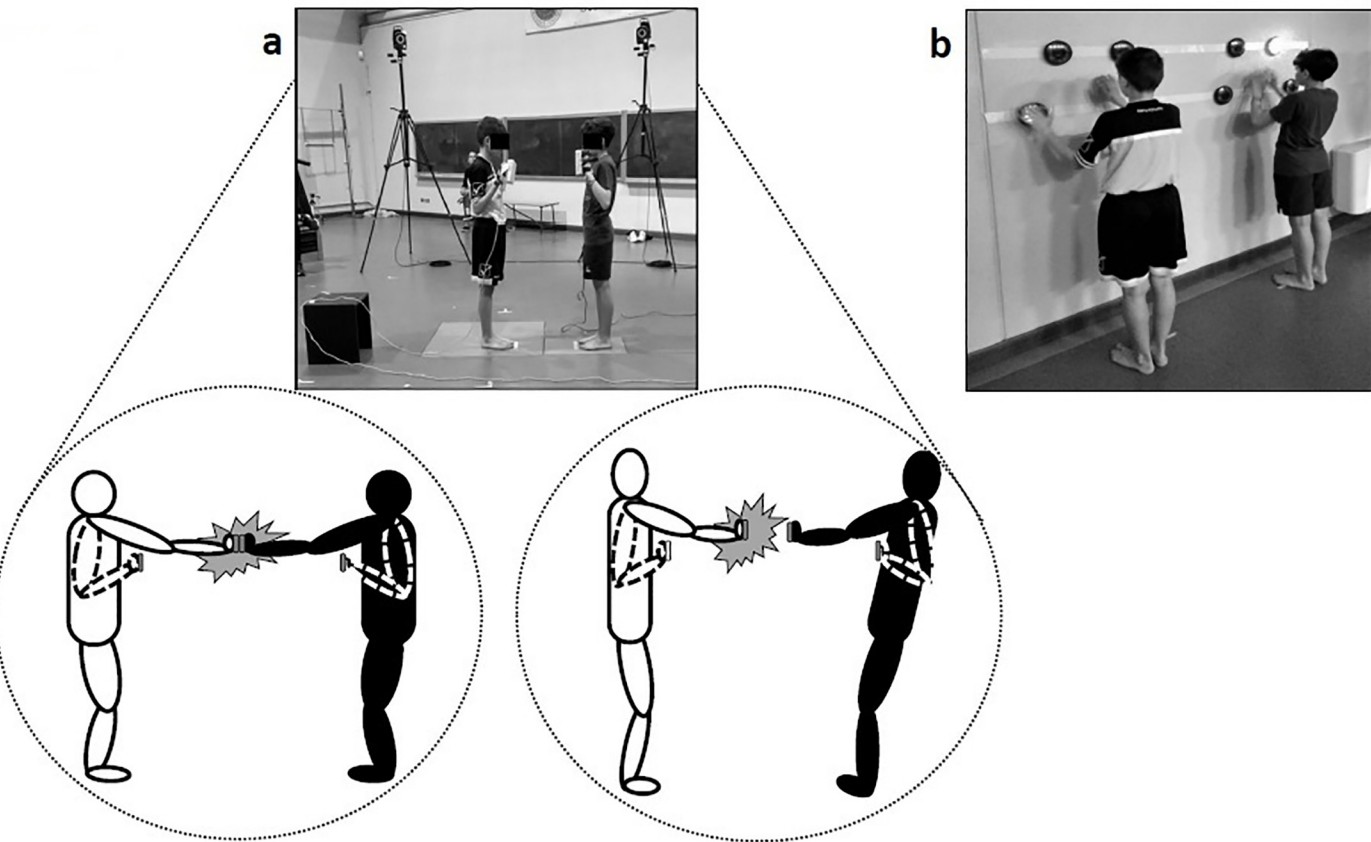

**Fig 1. The experimental set-up. a**. The proactive type of game (left diagram: the impact was associated with a perfect stability; right diagram: the impact was associated with a loss of stability). **b**. The reactive type of game.

signal but instead to self-select their own time of action initiation. The main aim of the game was inducing players to guess the opponent's action for anticipating and actuating the best movement's strategy (Fig 1A, right diagram). At the end of each trial the players were asked to regain the initial position, ready to initiate the next trial. For each trial the experimenter checked the posture taken by players to make sure that the rules were followed.

A total of 10 blocks with 12 trials each were performed for each couple of players; there was an interval between blocks of 3 minutes to avoid fatigue. A total of 960 trails were performed; from this total we selected, from each couple, the number of trials where the impact was present, obtaining a total of 560 trials considering all the couples together. To assure the maximum game farness the selection of players forming each couple was made based on the similarity in the anthropometrics measures in terms of body height and weight.

**Reactive type of game.** In order to test the reactive skills, a FITLIGHT Trainer (FILIGHT Sports Corp., Ontario, Canada) was used. This is a wireless system consisting of light sensors. The lights were attached to the wall forming two semicircles made each of 4 light sensors. The two semicircles were identical and positioned at 1meter distance from each other, within each semicircle each light was distant from each other by 20 cm. Each player was standing in front of one of the two semicircles that was positioned at the level of the player's shoulder high (Fig 1, panel b). Each player stood in front of the 4 lights at a comfortable distance. The rule of the game was: as soon as a light appeared on, shut it down as fast as possible by touching it, then one other light will appear on and again as fast as possible shut it down by touching it and so on. The time of each trial was made of 30 seconds, and four trials were administered for each couple of players. As performance measures we considered the time between lights shut down and the total number of lights shut down. Players were allowed to use both hands, the sequences of the lights on positions were random; players were performing individually but simultaneously and were placed at 1 metre distance from each other.

Prior to data collection, a period of familiarization was given to the players for both games (proactive and reactive). The whole experimental session lasted for about 60 min.

## Data analysis

For the proactive game video-clips were taken. All video clips were scrutinized to identify the presence of falls by indicating for each player the "stability index" considering as value 1 the perfect stability and value 0 loss of stability (e.g. loss of stability was considered when one of the two feet or both were detached from the force platform as illustrated in Fig 1A, right diagram). The percentage of falls for each player was also measured. Each video was analyzed twice by the same operator in order to confirm data reliability.

Acceleration data were processed offline using the MATLAB software (MathWorks, Natick, MA, USA), and the signal from the three coordinates of the accelerometer was integrated. The acceleration data were digitally low-pass filtered at 20 Hz using a fifth order, zero-lag Butterworth filter. Firstly, the instant at which the subject initiated the movement ($T_0$) was calculated when the magnitude of this integrated acceleration exceeded 3% of its peak absolute magnitude in that particular trial. These values were confirmed by visual inspection. Secondly, the instant of impact ($T_1$) was appreciated by considering the mean value of the time of peak acceleration between the soccer player and the swimmer (see Fig 2 which shows a representative trial of the anterior-posterior GRF for a player). Movement time was calculated as the time from $T_0$ to $T_1$. The difference between the $T_0$ of the soccer player and the $T_0$ of the swimmer was considered as time lag that defines the player that initiated the action as first.

Also, the force data were digitally low-pass filtered at 20 Hz using a fifth order, zero-lag Butterworth filter. Anticipatory Postural Adjustments (APAs) were calculated as the absolute

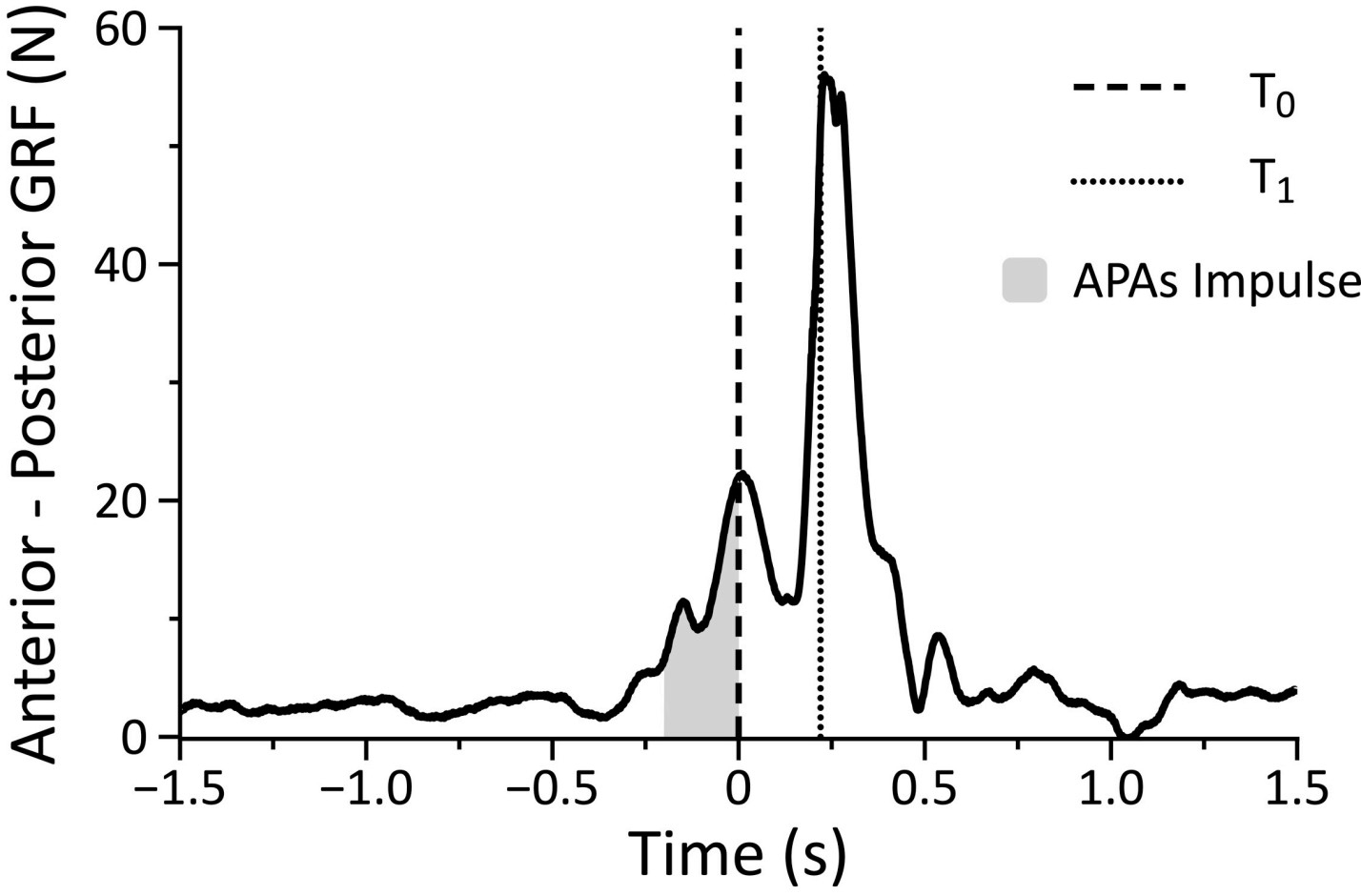

**Fig 2. A representative trial of the anterior-posterior GRF for a player.** $T_0$ is the instant at which the subject initiated the movement; $T_1$ is the instant of impact; absolute APAs impulse has been calculated as the integrated signal of the anterior-posterior GRF over a fixed window time (from -200 ms to $T_0$).

impulse generated by the anterior-posterior GRF over a window time from -200 ms to $T_0$. It is worth noting that, we only considered the anterior-posterior component of force because the movement took place mainly in such direction during the game. The fixed time of 200 ms was chosen based on previous APAs studies during fast voluntary movements [29, 35], as well as a priori visual inspection of the GRFs during the data analysis.

For the reactive game the number of lights off and the timing between each light off (reaction time expressed in seconds) were measured with the FITLIGHT Trainer.

## Statistical analysis

Results are presented as mean and standard deviation (SD). A Student t-test for paired data was performed to investigate group differences in anthropometrical and general characteristics ($p < 0.05$).

The normality of the distributions was assessed with the Kolmogorov-Smirnov goodness of fit test. For each variable (movement time, time lag, APAs impulse, and stability index), we performed a Mann-Whitney U-test for independent samples to investigate differences between soccer players and swimmers ($n = 8$; $p < 0.05$). Also, simple linear regression analysis was computed to investigate the correlations among all variables; the reliability (significance) of

**Table 2. Data studied for the proactive type of game and the reactive type of game (mean (SD)), and statistical significance between soccer players and swimmers.**

| Characteristic | Soccer players (n = 8) | 95% CI | Swimmers (n = 8) | 95% CI | Effect size (r) | p value |
|---|---|---|---|---|---|---|
| Proactive type of game | | | | | | |
| Movement Time (s) | 0.27 (0.04) | 0.24–0.31 | 0.30 (0.06) | 0.24–0.35 | -0.211 | p = 0.399 |
| Time Lag (s) | 0.11 (0.03) | 0.08–0.14 | 0.12 (0.02) | 0.09–0.14 | -0.160 | p = 0.522 |
| Trials with time lag (%) | 32.6 (14.6) | 18.0–47.2 | 37.4 (13.3) | 24.1–50.1 | -0.158 | p = 0.527 |
| APAs Impulse (N·s) | 2.34 (1.39) | 0.96–3.70 | 2.16 (1.01) | 1.15–3.20 | -0.053 | p = 0.834 |
| Stability Index | 0.87 (0.09) | 0.77–0.96 | 0.66 (0.11) | 0.55–0.77 | -0.709 | p < 0.01 |
| Reactive type of game | | | | | | |
| Reaction Time (s) | 0.45 (0.05) | 0.41–0.50 | 0.48 (0.07) | 0.41–0.55 | -0.263 | p = 0.294 |
| Number of lights off | 39.3 (2.18) | 37.1–41.5 | 38.2 (3.52) | 34.7–41.7 | -0.276 | p = 0.270 |

these relationships was established based on the coefficient of correlation (r) and the appropriate degrees of freedom (n-2) (e.g. [36]). The level of significance was set at p < 0.05. Statistical analysis was performed using SPSS program (version 22 for Windows).

## Results

There were no significant differences between the two groups in mean age, height, weight and years of experience (see Table 1) (Student t-test for paired data; p > 0.05).

Average and standard deviation values measured in both proactive and reactive game are reported in Table 2 for soccer players and swimmers (Mann-Whitney U-test for independent samples; p < 0.05).

In the proactive game, the time lag (mean ± SD of soccer players and swimmers: 0.11 ± 0.03 and 0.12 ± 0.02 s; p = 0.522), the movement time (0.27 ± 0.04 and 0.30 ± 0.06 s; p = 0.399) and APAs impulse (2.34 ± 1.39 and 2.16 ± 1.01 N·s; p = 0.834) were not significantly different between the two groups.

Importantly, though soccer players were significantly more stable than swimmers presenting a higher index of stability (0.87 ± 0.09 and 0.66 ± 0.11; p < 0.01; mean ± SD of soccer players and swimmers) and a significantly less percent of falls (13.4 ± 9.5 and 34.4 ± 11.0%; p < 0.01, respectively) (Fig 3, panel a). Moreover, just soccer players, presented a significant linear correlation between APAs impulses and time lag (APAs Impulse = -1.211–32.929·time lag for soccer players: r = 0.751; p < 0.01) while there was none correlation for swimmers (APAs Impulse = 3.218–9.024·time lag for swimmers: r = 0.205; p = ns). (Fig 3, panel b). No other significant correlations were found for the proactive game.

As for the reactive game, the statistical analysis showed none significant difference between soccer players and swimmers: all players turned off a similar number of lights (39.3 ± 2.18 and 38.2 ± 3.52; p = 0.270) by applying the same time (0.45 ± 0.05 and 0.48 ± 0.07 s; p = 0.294).

## Discussion

Our study was designed to investigate the development of anticipatory and strategic skills in total body games competitions comparing young expert soccer players (open skill-dominated sport) with swimmers (closed skill-dominated sport) of the same age and level of sport expertise.

In particular, we aimed at disentangling proactive and reactive control mechanisms in individuals trained in different sports as close and open skills related. In general, we found that while there were none differences between swimmers and soccer players in their ability to apply reactive control, soccer players outperformed swimmers when engaged in a competition where a proactive control was required. Importantly results showed that the superior ability

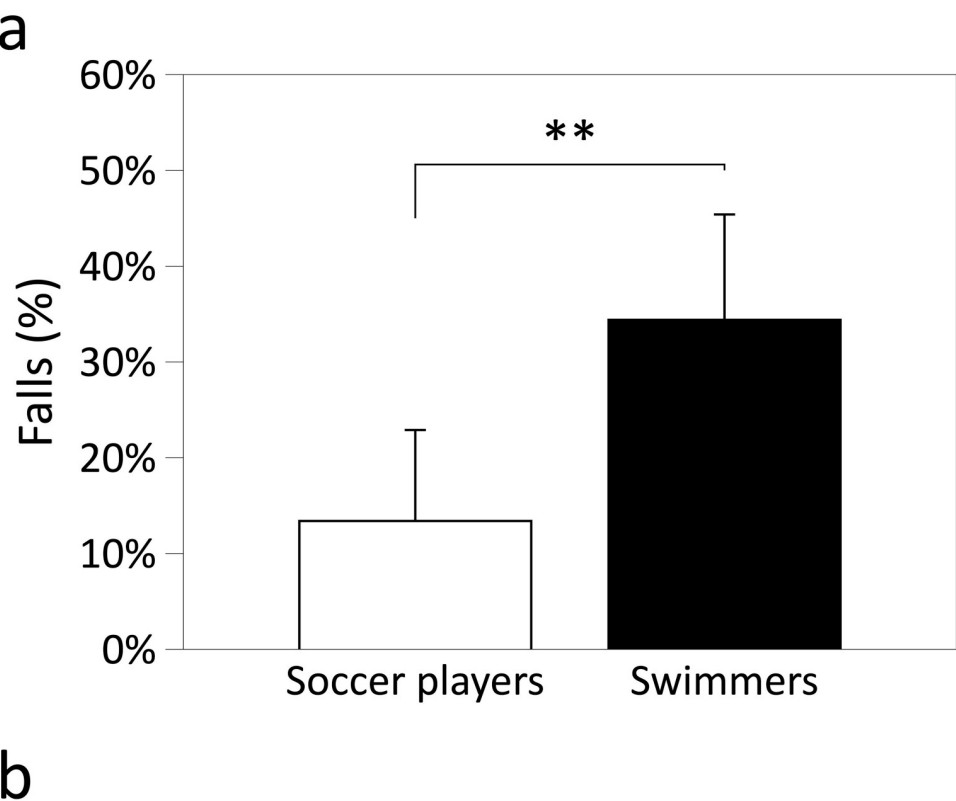

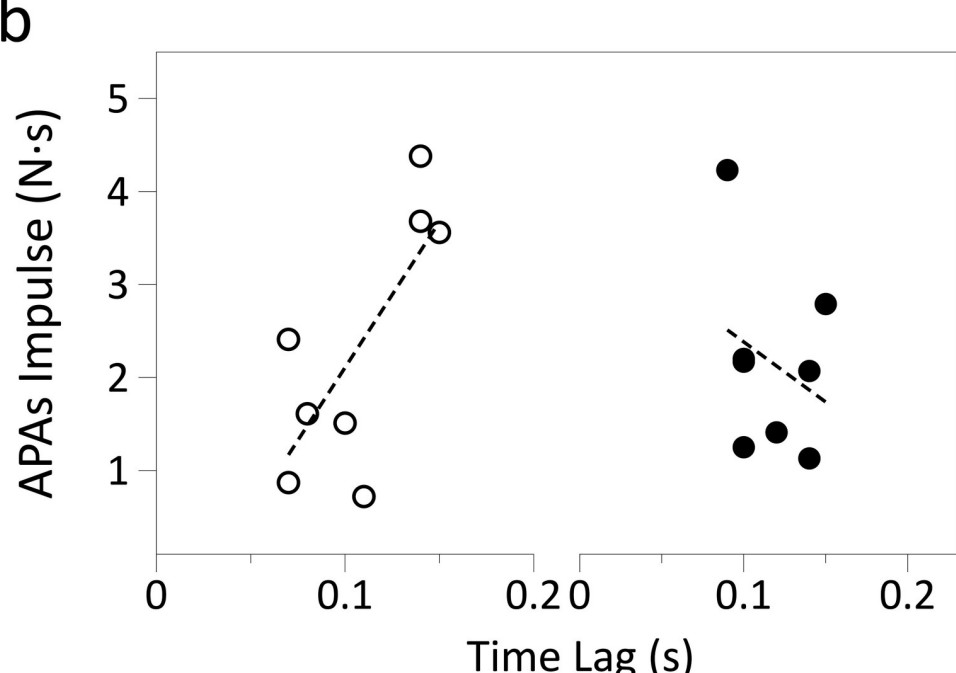

**Fig 3. Proactive game's findings. a**. Percentage of falls of soccer players (white bars) and swimmers (black bars) for the proactive type of game. Average values ± SD have been reported, as well as statistical significance. **b**. APAs Impulse is plotted as a function of time lag for the proactive type of game. The symbols refer to: soccer players ○, and swimmers ●.

presented by soccer players was ascribed to the proper modulation of Anticipatory Postural Adjustments in preparation to their attack to the opponent. These findings underline the specific development of skills abilities due to the peculiarity of a sport training.

Our results were consistent with previous studies [33], in defining skills/environment relationship where in a highly predictable environment closed skills are required as a prompt reaction given a known signal, while in the unpredictable environment athletes are forced to anticipate impending events by using open skills. In swimming, the sound of the starter's gun indicates the race initiation and a fast reaction to auditory stimuli is required; a slight delay in reacting to the gun sound might change the overall result in a competition [2, 4]. On the other hand, soccer players have been trained in a team and the game is performed in a highly dynamic environment in which athletes continuously predict the ball and the opponents position now and sooner after now [37].

Here we showed that during a typical proactive competition, soccer players were able to beat the swimmers by maintaining higher stability and falling a significantly smaller number of times. Importantly, we showed that the main "weapon" soccer players used was to activate a proper action anticipation by modulating their APAs. APAs were arranged based on the amount of body impulse impinged on the force platform before action initiation and this amount of force was a function of the time gap between the action initiation of the soccer player with respect to the action initiation of the opponent. Higher impulse was impinged for more anticipated actions and lower impulse for less anticipated action, and this impulse modulation was present only for soccer players. We sustain that this was the strategy that allowed a soccer player to obtain a greater postural stability at the instant of the impact with the body's swimmer and, as a consequence, to win the game. In other words, soccer players were able to modulate their APAs based on their force/timing relationship impinged in their body preparation defined by their prediction of future movement initiation of the opponent player [5, 37].

Our results are sustained by researches showing that open skill related athletes outperform closed skills related athletes in the proactive contests [38], but here we added the relevant information of the feedforward motor command from the motor cortex that sustains the development of these type of skills. Some researches posed the question whether results can be obtained only when the task to be performed is specific of the sport in which the athletes excel [17]. Here, we applied a game competition that involved actions that were not specific for soccer nor for swim sport. This was important for showing that the proactive type of control is independent from the type of movement performed [17].

In line with previous works [39], as in our research, reaction time has been evaluated by means of an oculo-manual coordination task using the FITLIGHT Trainer system (Sports Corp., Ontario, Canada). In this case the game was mainly based on reactive control including the competition since the two players were performing the game at the same time and they could have the sense of the other's performance. But in this case, we found similar performance for soccer players and swimmers, suggesting that all the recruited athletes presented a high physical condition, promptness, ability to face stressful situations and great concentration [3]. Furthermore, all our subjects could execute likewise the mental operations needed by the task at hand, with the same speed of processing, showing an equal index of efficiency [40].

We believe that the results from our study are unlikely confounded by the consistency of the sample gender. Future studies should be done considering also a female gender sample.

## Limitations

This paper presents a number of limitations. We recruited a limited number of participants, 16 athletes, but this was partially compensated by a large number of trials performed by each

participant: at least 65 trials for each couple of athletes. In future works, it would be interesting to involve a larger number of participants and testing different levels of sport experience [41] and following closely the development of reactive and proactive control ability as children grow and acquire over time specialized training. An interesting additional component that could be considered in future experiment will be eye and head movement to detect more finely the anticipated intention of the player and or possible subtle fake strategies. This issue could be addressed in future by employing some specific technologies such as the electronic goggles and a head movement controller.

## Supporting information

**S1 Table. Normality test.**
(PDF)

**S1 Data. File data.**
(XLSX)

## Acknowledgments

We would like to thank Edoardo Ribani and Enrico Salgarollo for their help in data collection and analysis, and the subjects for participating the study.

## Author Contributions

**Conceptualization:** Francesca Nardello, Paola Cesari.

**Data curation:** Francesca Nardello, Matteo Bertucco, Paola Cesari.

**Formal analysis:** Francesca Nardello, Matteo Bertucco, Paola Cesari.

**Funding acquisition:** Francesca Nardello, Paola Cesari.

**Investigation:** Francesca Nardello, Matteo Bertucco, Paola Cesari.

**Methodology:** Francesca Nardello, Matteo Bertucco, Paola Cesari.

**Project administration:** Francesca Nardello, Paola Cesari.

**Resources:** Francesca Nardello, Matteo Bertucco, Paola Cesari.

**Software:** Francesca Nardello, Matteo Bertucco, Paola Cesari.

**Supervision:** Francesca Nardello, Paola Cesari.

**Validation:** Francesca Nardello, Matteo Bertucco, Paola Cesari.

**Visualization:** Francesca Nardello, Matteo Bertucco, Paola Cesari.

**Writing – original draft:** Francesca Nardello, Matteo Bertucco, Paola Cesari.

**Writing – review & editing:** Francesca Nardello, Matteo Bertucco, Paola Cesari.

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
