## [Decision Letter · Decision Letter 0]

19 Jan 2021

PONE-D-20-36591

Anticipatory and pre-planned actions: a comparison between young soccer players and swimmers

PLOS ONE

Dear Dr. Cesari,

Thank you for submitting your manuscript to PLOS ONE. After careful consideration, we feel that it has merit but does not fully meet PLOS ONE’s publication criteria as it currently stands. Therefore, we invite you to submit a revised version of the manuscript that addresses the points raised during the review process.

The reviewer who has evaluated your MS has judged it interesting and well conducted. A number of suggestions and remarks however should be addressed before endorsing its publication. I've read the MS and agree with the reviewer's remarks. Please also double check language use for amending some errors.

We look forward to receiving your revised manuscript.

Kind regards,

Cosimo Urgesi

Academic Editor

PLOS ONE

Journal Requirements:

2.Thank you for stating the following financial disclosure:

 "NO - The funders had no role in study design, data collection and analysis, decision to publish, or preparation of the manuscript."

3. Please state in your methods section whether you obtained consent from parents or guardians of the minors included in the study or whether the research ethics committee or IRB approved the lack of parent or guardian consent.

Reviewers' comments:

Reviewer's Responses to Questions

**Comments to the Author**

1. Is the manuscript technically sound, and do the data support the conclusions?

Reviewer #1: Yes

2. Has the statistical analysis been performed appropriately and rigorously? 

Reviewer #1: Yes

3. Have the authors made all data underlying the findings in their manuscript fully available?

Reviewer #1: Yes

4. Is the manuscript presented in an intelligible fashion and written in standard English?

Reviewer #1: Yes

5. Review Comments to the Author

Reviewer #1: Dear Authors,

Thank you for the opportunity to revise your manuscript.

General comments:

This study aimed to investigate anticipatory and pre-planned actions in young athletes (eight soccer players and eight swimmers). Although the study targets an important topic for better understating the difference in reactive and proactive control for sport considering open and closed skills, several shortcomings should be addressed to improve the readability and reproducibility of the manuscript.

Major comments

Introduction

• While the Introduction appears to be sound, the structure is not so clear, making it difficult to follow. I advise the Authors to improve the flow and readability of the text. For example, the Aims should be at the end of the Introduction paragraph.

• Authors should point out that Anticipatory Postural Adjustments (APAs) are usually studied with EMG and/or with centre of pressure displacement. Usally APAs are seen prior to an expected postural perturbation as early EMG changes [reviewed in Belenkiy et al. 1967; Massion 1992].

• Add a small paragraph about using the Ground Reaction forces when investigating APAs.

• Authors should clearly state the underlying hypotheses for developing the study. The hypotheses should be linked with the objectives.

Material and methods

• Please add the statistical tests used and the P values for investigating differences between Participants.

• Why were no females recruited? This should be discussed.

• Data Analysis:

The stability index was defined according to a dichotomous scale 0 and 1. Why did the Authors not use the data from the force platform (e.g., COP displacement) to determine the loss of stability?

Moreover, the same operator that analyzed the video, may carry a systematic error.

APAs data processing should be more detailed. How the GRFs ground reaction force components (Fz,Fx, and Fz) were computed?. How was the data filtered?

There is no mention of durations, amplitudes, and impulses of the reaction forces.

For the anterior-posterior GRF was set a threshold value to achieve? How were these values determined? For example, the anterior-posterior GRFs exceeded ±2 SD from its mean computed across trials in one session for each participant.

How were APAs impulse defined? It is unclear the measurement units of APAs.

Results

• Authors should add plots of the anterior-posterior GRF, at least for one subject (e.g., average across trials). Readers will appreciate the time series for the APA impulses.

• Please add in the Results section the statistical test used, with the test values and the degrees of freedom.

• The effect size reported should be the (r) for the Mann-Whitney U-test that is Effect size: r ( = Z/(√Nobs)Z/(Nobs)). Here the Cohen’s d is reported, thus overestimating the effect.

• The small sample size might bias the correlation based on the coefficient of correlation (r). I wonder if robust correlation methods (e.g., bootstrapping, data winsorizing, skipped correlations) or the Spearman rank correlation should be applied.

Discussion

The discussion is well presented.

Minor Comments

In the abstract should be stated the age (mean and SD) of the participants. Add how the APAs were measured, e.g., through the force platform. It may be interesting to add the APAs values.

Once an abbreviation has been presented, it is not necessary to repeat it, e.g., Line 125, the ground reaction force (GRF)

6. PLOS authors have the option to publish the peer review history of their article (what does this mean?). If published, this will include your full peer review and any attached files.

Reviewer #1: No

---

## [Author Response · Author response to Decision Letter 0]

19 Feb 2021

Comments from the Editor and Reviewers

We would like to thank the reviewer for his/her helpful comments and for the opportunity to revise our manuscript. We believe that, by addressing the Reviewer’s concerns, the revised manuscript has significantly improved.

Major changes are:

• the data analysis when investigating APAs has been supplemented with further details, as requested. A new figure (Figure 2 in this revised manuscript) has been reported;

• related references are added;

Changes introduced in the manuscript are enlightened in yellow.

Following we will consider the comments point by point. The modifications to the text will be reported in italics.

Reviewer #1

Major comments

Introduction

While the Introduction appears to be sound, the structure is not so clear, making it difficult to follow. I advise the Authors to improve the flow and readability of the text. For example, the Aims should be at the end of the Introduction paragraph.

ANSWER. As suggested, the change has been done by adding the explanation of the main aim of the study at the end of the introduction.

“[…] The main aim of this study is to shade light on the role of skills development (open and closed) by comparing APAs, by means of the ground reaction forces, in children highly experienced in open and closed disciplines respectively soccer and swim […]”.

Authors should point out that Anticipatory Postural Adjustments (APAs) are usually studied with EMG and/or with center of pressure displacement. Usually APAs are seen prior to an expected postural perturbation as early EMG changes [reviewed in Belenkiy et al. 1967; Massion 1992].

ANSWER. We agree with the point raised by the Reviewer. We are aware of EMG (see references 22,25), and ground reaction forces (see references 24,26-27) and COP displacement (see reference 28) for investigating the Anticipatory Postural Adjustments (APAs). As required, these concepts are explained in more details in the revised version of the manuscript.

“In literature, APAs are usually studied with EMG analysis (i.e. early postural adjustments, EPA) [22-25] and/or with force platform [24,26-28]. APAs are typically quantified in terms of their magnitude or the onset timing of the signal (i.e. EMG or GRFs) before the initiation of the movement [26,27]. When measured with the ground reaction force [29-31], APAs onset, amplitude and impulse are usually analysed from the ground reaction forces (GRFs) [24,26-,27] or center of pressure [28]”.

Add a small paragraph about using the Ground Reaction Forces when investigating APAs.

ANSWER. A small paragraph has been inserted (see also the comment above) and related references added (see references 29-31).

[29] Aruin AS, Latash ML. Directional specificity of postural muscles in feed-forward postural reactions during fast voluntary arm movements. Exp Brain Res.1995;103: 323-332.

[30] Yamagata M, Gruben K, Falaki A, Ochs WL, Latash ML. Biomechanics of Vertical Posture and Control with Referent Joint Configurations. Journal of Motor Behavior. 2020;53(5): 1-11.

[31] Yamagata M, Falaki A, Latash ML. Stability of vertical posture explored with unexpected mechanical perturbations: synergy indices and motor equivalence. Experimental Brain Research. 2018;236(2). Doi: 10.1007/s00221-018-5239-x.

Authors should clearly state the underlying hypotheses for developing the study. The hypotheses should be linked with the objectives.

ANSWER. The integration has been made as suggested.

“In a proactive game, we would expect a greater postural stability accomplished with a lower risk of falling for the open-skills players in comparison to the closed-skills ones. This augmented ability would be reflected by an early modulation of the Anticipatory Postural Adjustments. In a reactive game, instead, we assume no differences between open and closed skills”.

Material and methods

Please add the statistical tests used and the P values for investigating differences between participants.

ANSWER. We have made the changes as suggested.

“[…] A Student t-test for paired data was performed to investigate group differences in anthropometrical and general characteristics (p < 0.05). […]. For each variable (movement time, time lag, APAs impulse, and stability index), we performed a Mann-Whitney U-test for independent samples to investigate differences between soccer players and swimmers (n = 8; p < 0.05) […]”.

Why were no females recruited? This should be discussed.

ANSWER. We recruit male subjects mainly for a reason. Given the nature of the task, it was required to match along with the level of sport experience also the anthropometric measures in terms of body height and weight since one of the crucial factors of the proactive game was the “homogeneity” between players. It was difficult to find adolescents’ females that matched the anthropometric measures for the two sport disciplines, and particularly due to the difficulty to recruit highly experienced females playing soccer.

“We believe that the results from our study are unlikely confounded by the consistency of the sample gender. Future studies should be done considering also a female gender sample.”.

Data Analysis

The stability index was defined according to a dichotomous scale 0 and 1. Why did the Authors not use the data from the force platform (e.g., COP displacement) to determine the loss of stability?

ANSWER. The stability index was defined according to the displacement of at least one foot from the floor. Importantly such displacement can only be detected by a video analysis. The COP migration instead, does not capture such changes, considering that most of the time the feet’ detachments from the floor were of a very small amplitude.

The videos were checked by one operator by applying a frame-by-frame analysis of the movement kinematics. The task for the operator was to define the presence of the displacement of at least one foot from the floor, and this operation did not include any bias given by a subjective interpretation; in order to assure more careful precision since the video checking was performed “manually” we performed the analysis twice.

APAs data processing should be more detailed. How the GRFs ground reaction force components (Fz, Fx, and Fz) were computed? How was the data filtered?

ANSWER. A priori analysis of the anticipatory postural phase showed a no monotonic function of the anterior-posterior GRFs (GRFx). Therefore, we opted for computing the absolute impulse generated by this force to quantify the magnitude of the APAs. This parameter was calculated as the integral of the anterior-posterior over the 200 ms before the initiation of the movement. It is worth noting that, we only considered the anterior-posterior component of force because the movement took place mainly in such direction during the game.

These force data were low pass filtered at 20 Hz by using a fifth-order, zero-lag Butterworth filter (same as for the acceleration data).

“Two force platforms (model OR-5, AMTI, USA: 90 x 90 cm and model Kistler, Switzerland: 40 x 60 cm) were used to record the three components of the ground reaction forces with a sampling rate of 2000 Hz. Only the anterior-posterior component (GRFx) has been used for further analysis”.

There is no mention of durations, amplitudes, and impulses of the reaction forces.

ANSWER. One of the proactive game’s rules was that the players were free to decide when to initiate their action; therefore, the movement onset was not standardized across trials. Differently than works investigating APAs, in this experiment the onset was defined by the diverse situations occurred during the game, as for instance: i) the player moved before the opponent player (i.e. anticipation); ii) the player moved in response to the movement of the opponent player; iii) the two players moved quite simultaneously. To better clarify the task, we added the timing information.

“After the push command players were free to decide when to initiate the action. No constraints for time initiation of the movement was given”.

Since the game was initiated autonomously by each player, the definition of the time onset was not consistent across trials. As a consequence, we considered to integrate the anterior-posterior force data in a window time defined between the initiation of movement (T0) and 200 ms before this instant. This procedure has been already proved in previous studies (see references 29 and 35); we included related information in the text.

[35] Aruin AS, Latash ML (1996) Anticipatory postural adjustments during self-initiated perturbations of different magnitude triggered by a standard motor action. Electroencephalogr Clin Neurophysiol. 101:497-503.

Therefore, our impulse corresponds to the integrated force multiplied by the window time considered (measurement unit of Newton * second).

More information about this have been inserted in the Data Analysis section.

“Also, the force data were digitally low-pass filtered at 20 Hz using a fifth order, zero-lag Butterworth filter. Anticipatory Postural Adjustments (APAs) were calculated as the absolute impulse generated by the anterior-posterior GRF over a window time from -200 ms to T0. It is worth noting that, we only considered the anterior-posterior component of force because the movement took place mainly in such direction during the game. The fixed time of 200 ms was chosen based on previous APAs studies during fast voluntary movements [29,35], as well as a priori visual inspection of the GRFs during the data analysis”.

For the anterior-posterior GRF was set a threshold value to achieve? How were these values determined? For example, the anterior-posterior GRFs exceeded ± 2 SD from its mean computed across trials in one session for each participant.

ANSWER. No threshold parameters were taken into consideration. All trials were considered for the analysis. We added this information in the text (see also the comment above).

How were APAs impulse defined? It is unclear the measurement units of APAs.

ANSWER. We calculate the amount (i.e. amplitude) of the anterior-posterior force considering the window time (from -200 ms to T0). The measurement units of such impulse was then Newton * second. We added more detailed explanations in the text (see also the comment above).

Results

Authors should add plots of the anterior-posterior GRF, at least for one subject (e.g., average across trials). Readers will appreciate the time series for the APAs impulses.

ANSWER. We thank for the suggestion, and to appreciate the time series for the APAs impulses we present a (new) Figure 2 showing one representative trial of the anterior-posterior GRF (we considered not pertinent presenting the average data across trials due to the presence of variability linked to the different strategies taken during the game).

“Fig 2. A representative trial of the anterior-posterior GRF.

T0 is the instant at which the subject initiated the movement; T1 is the instant of impact; absolute APAs impulse has been calculated as the integrated signal of the anterior-posterior GRF over a fixed window time (from -200 ms to T0)”.

Please add in the Results section the statistical test used, with the test values and the degrees of freedom.

ANSWER. Both the Statistical Analysis and the Result sections have been modified by following this suggestion. As there are no parameter values being estimated from the data in the non-parametric Mann-Whitney test, it doesn’t really involve degrees of freedom in the same way as t-tests and other parametric tests do. The number of cases and means have been reported for each group.

“There were no significant differences between the two groups in mean age, height, weight and years of experience (see Table 1) (Student t-test for paired data; p < 0.05).

Average and standard deviation values measured in both proactive and reactive game are reported in Table 2 for soccer players and swimmers (Mann-Whitney U-test for independent samples; p < 0.05)”.

The effect size reported should be the (r) for the Mann-Whitney U-test that is Effect size: r (= Z/(√Nobs)Z/(Nobs)). Here the Cohen’s d is reported, thus overestimating the effect.

ANSWER. As kindly suggested, in Table 2 it has now been reported the (r) values for the Mann-Whitney U-test.

The small sample size might bias the correlation based on the coefficient of correlation (r). I wonder if robust correlation methods (e.g., bootstrapping, data winsorizing, skipped correlations) or the Spearman rank correlation should be applied.

The decision to apply the Pearson correlation instead of Spearman rank correlation was taken for these reasons:

1- Pearson correlation is applicable for a normal distribution while Spearman is suitable for non-normal distribution. The data in this experiment follow a normal distribution.

2- When data are normally distributed Pearson correlation is still suggested even in presence of a small sample size.

For more details, see Kowalski CT. On the Effects of Non-Normality on the Distribution of the Sample Product-Moment Correlation Coefficient. Journal of the Royal Statistical Society. Series C (Applied Statistics), 1972;21(1): 1-12.

Discussion

The discussion is well presented.

ANSWER. Thank you for this positive comment.

Minor Comments

In the abstract should be stated the age (mean and SD) of the participants. Add how the APAs were measured, e.g., through the force platform.

ANSWER. The age of the participants has now been reported, as well as how the APAs were measured.

“[…] Sixteen young (11-12 years) athletes (eight soccer players and eight swimmers). […] By means of kinematic (i.e. movement time and duration) and dynamic analysis through the force platform (i.e. Anticipatory Postural Adjustments, APAs) […]”.

It may be interesting to add the APAs values.

ANSWER. We prefer not to insert these values in the abstract, but just to report them in the text (and in Table 2).

Once an abbreviation has been presented, it is not necessary to repeat it, e.g., Line 125, the ground reaction force (GRF).

ANSWER. S

---

## [Decision Letter · Decision Letter 1]

17 Mar 2021

PONE-D-20-36591R1

Anticipatory and pre-planned actions: a comparison between young soccer players and swimmers

PLOS ONE

Dear Dr. Cesari,

Thank you for submitting your manuscript to PLOS ONE. After careful consideration, we feel that it has merit but does not fully meet PLOS ONE’s publication criteria as it currently stands. Therefore, we invite you to submit a revised version of the manuscript that addresses the points raised during the review process.

Please address the two minor revisions requested by the reviewer before I can endorse publication.

We look forward to receiving your revised manuscript.

Kind regards,

Cosimo Urgesi

Academic Editor

PLOS ONE

Journal Requirements:

Reviewers' comments:

Reviewer's Responses to Questions

**Comments to the Author**

1. If the authors have adequately addressed your comments raised in a previous round of review and you feel that this manuscript is now acceptable for publication, you may indicate that here to bypass the “Comments to the Author” section, enter your conflict of interest statement in the “Confidential to Editor” section, and submit your "Accept" recommendation.

Reviewer #1: All comments have been addressed

2. Is the manuscript technically sound, and do the data support the conclusions?

Reviewer #1: Yes

3. Has the statistical analysis been performed appropriately and rigorously? 

Reviewer #1: Yes

4. Have the authors made all data underlying the findings in their manuscript fully available?

Reviewer #1: Yes

5. Is the manuscript presented in an intelligible fashion and written in standard English?

Reviewer #1: Yes

6. Review Comments to the Author

Reviewer #1: Dear Authors,

Thank you for the opportunity to revise your Manuscript.

I appreciate your changes made on the paper.

I do not have any major Comments.

Minor Comments:

Line 247 – 248. I think there is a typo error. P values should be p > 0.05 as no significant differences were detected in between the two groups in mean age, height, weight and years of of experience.

Pleas add a note to (r) in Table 2 as Effect size. It may be misleading as later r is the correlation coefficient.

7. PLOS authors have the option to publish the peer review history of their article (what does this mean?). If published, this will include your full peer review and any attached files.

Reviewer #1: No

---

## [Author Response · Author response to Decision Letter 1]

17 Mar 2021

Reviewer #1

Dear Authors,

Thank you for the opportunity to revise your Manuscript.

I appreciate your changes made on the paper.

I do not have any major Comments.

ANSWER. Thank you very much for your positive comment.

Minor Comments

Line 247 - 248. I think there is a typo error. P values should be p > 0.05 as no significant differences were detected in between the two groups in mean age, height, weight and years of experience.

ANSWER. This was a typing mistake: sorry about this.

Please add a note to (r) in Table 2 as Effect size. It may be misleading as later r is the correlation coefficient.

ANSWER. The integration has been made as suggested.

---

## [Editor Report · Decision Letter 2]

23 Mar 2021

Anticipatory and pre-planned actions: a comparison between young soccer players and swimmers

PONE-D-20-36591R2

Dear Dr. Cesari,

We’re pleased to inform you that your manuscript has been judged scientifically suitable for publication and will be formally accepted for publication once it meets all outstanding technical requirements.

Kind regards,

Cosimo Urgesi

Academic Editor

PLOS ONE
---

## [Editor Report · Acceptance letter]

25 Mar 2021

PONE-D-20-36591R2 

Anticipatory and pre-planned actions:a comparison between young soccer players and swimmers 

Dear Dr. Cesari:

I'm pleased to inform you that your manuscript has been deemed suitable for publication in PLOS ONE. Congratulations! Your manuscript is now with our production department. 

Kind regards, 

on behalf of

Dr. Cosimo Urgesi 

Academic Editor

PLOS ONE